# B-Spline Curve Fitting of Hungry Predation Optimization on Ship Line Design

**Changle Sun \*, Mingzhi Liu and Shihao Ge**

Department of Mechanical Engineering, Dalian Maritime University, Dalian 116026, China
* Correspondence: sunchangle@dlmu.edu.cn

**Abstract:** The ship line often describes by the offset table of discrete data points, which leads to the problems that three view coordinates may not correspond, the fitting error is large and the fitted curve cannot be easily modified. This will seriously affect the subsequent ship performance evaluation and op-timization. To solve this problem, this paper develops a B-spline curve fitting of hunger predation optimization on ship line design (HPA), which contains knot guidance technology, hungry preda-tion optimization technology and adaptive adjustment of algorithm input parameters. HPA transforms the discrete ship line into a continuous B-spline curve description, which improves the accuracy and modifiability of the ship line design. Through the real-time feedback of the results of each round of iteration, the knot vector is adaptively adjusted towards a better fitness, and then the optimal control point set that satisfies the error threshold can be obtained. The effectiveness and superiority of HPA are verified by comparing with related research and engineering software.

**Keywords:** knot placement; b-spline curve fitting; ship line design; optimization algorithm; reverse engineering

## 1. Introduction

B-spline curve approximation has been a hot research topic in the field of computer-aided geometric design in recent years. With the development of computer-aided design (CAD) and the rise of advanced manufacturing technology, B-spline is widely used in the design and manufacturing industry of free curves and surfaces [1–5]. In the field of shipbuilding, the design of ship lines is the basis of shipbuilding [6]. The ship lines are closely related to the dynamic characteristics and resistance characteristics of the ship. Improving the design level of ship lines and then improving the design quality of ships has become a hot research topic in the field of shipbuilding industry [7,8]. However, due to the error in the design of the ship lines or the modification in the local smoothing process, the coordinates of the corresponding points of the three views may not correspond, resulting in the reduction of the accuracy of the ship line diagram [9]. In addition, the data points in the ship line diagram are discrete, and two adjacent discrete points are connected by straight line segments, resulting in poor smoothness, which brings trouble to the subsequent ship line setting out and processing and affects the quality and efficiency of shipbuilding.

Due to the locality of B-spline, it can be more convenient to modify the curve locally. The parameter continuity of B-spline can ensure that the fitting curve has good smoothness. If the continuous B-spline method is used in the ship line design to replace the discrete offset table method, the accuracy of ship line design will be greatly improved, and the subsequent multi-disciplinary collaborative optimization and manufacturing will be facilitated.

Compared with interpolation, B-spline curve approximation can better reflect the shape of curve, but the placement of knot vector is still an urgent problem to be solved. Unreasonable knot vector arrangement may lead to unacceptable shapes [10]. Researchers put higher fitting accuracy at the same knot number [11,12], fewer knot number under the same accuracy requirements [13,14] and faster operating efficiency [15,16] as the main pursuit goals of B-spline curve fitting.

This paper develops a B-spline curve fitting of hunger predation optimization on ship line design (HPA). In the previous dynamic knot method [17–22], the initial knots are randomly selected from the interval [0, 1], so there are many unreasonable knot vectors. Therefore, large population size and iteration number are required, which leads to low efficiency of the algorithm. The knot guidance technology is designed to add knots in the area with complex model shape at the initial knot selection stage. The premature loss of population diversity in optimization algorithm leads to slow convergence and is easy to fall into local optimality. A hunger search strategy is developed to make the hungry individuals in the population approach the optimal solution more quickly, and the influence of neighbors on the position adjustment is further considered. Aiming at the problem that the previous dynamic knot method requires manual adjustment of key input parameters, such as population size and iteration number, which is troublesome and time consuming. An adaptive adjustment of key input parameters in the HPA algorithm is proposed, which can quickly adapt to the replacement of model and fitting accuracy. HPA achieves the goal of B-spline curve fitting with higher fitting accuracy at the same control points, less control points under the same accuracy requirements, faster operation efficiency and better universality; it can better solve practical engineering problems.

The rest of the paper is organized as follows. Section 2 introduces the basic B-spline theory used in HPA algorithm and research on existing B-spline curve fitting. Section 3 proposes the HPA method, including a new knot guidance technology, a hungry predation optimization technology, fitness function selection and the dimension calculation rule. Section 4 analyzes the influence of population size, iteration number and error threshold of HPA on curve fitting results, and an adaptive adjustment method of initial input parameters is proposed. Section 5 verifies the performance of HPA algorithm; we verify the superior performance of HPA algorithm in fitting accuracy and convergence speed by comparing with typical static knot method, dynamic knot method and engineering commercial software. Finally, we summarize this paper and look forward to future research directions.

## 2. Related Works

### 2.1. B-Spline Theory Knowledge

This section refers to the bibliography [23,24]. The knot vector $U$ consists of the non-decreasing parameter $u_i$, $U$: $u_0 \leq u_1 \leq \cdots \leq u_{n+\rho+1}$, where the dimensions and values of the initial knot vector are random under certain constraints. After that, they will be adjusted with each iteration. It will be introduced in detail in Section 3. So far, B-spline basis function $N_{i,\rho}(x)$ can be solved, as shown in Formula (1) and (2),

$$N_{i,0}(x) = 1, \quad u_i \leq x \leq u_{i+1} \ (i = 0, \ldots, n)$$
$$0, \quad \text{other}$$
(1)

$$N_{i,\rho}(x) = \frac{x - u_i}{u_{i+\rho} - u_i} N_{i,\rho-1}(x) + \frac{u_{i+\rho+1} - x}{u_{i+\rho+1} - u_{i+1}} N_{i+1,\rho-1}(x)$$
(2)

where the first subscript $i$ represents the sequence number, and the second subscript $\rho$ represents degree (order $\rho$ + 1) of curve.

Inappropriate degree may lead to the fitting curve cannot meet the accuracy requirement. Designers prefer to use piecewise low degree curves to describe complex curves, and they do not want a curve to be divided into too many segments. Because the cubic curve is not only a plane curve with inflection point, but also the lowest degree of spatial curves, it is widely used [24].

A polynomial B-spline curve of degree $\rho$ (or order $\rho$ + 1) is a piecewise polynomial curve given by

$$p(x) = \sum_{i=0}^{n} d_i N_{i,\rho}(x) \quad x \in [0, 1]$$
(3)

where $d_i$ is the control point of the curve and $N_{i,\rho}(x)$ is the $\rho$-degree-normalized B-spline basis function.

According to Formula (3), it is necessary to obtain the control point $d_i$ to realize the curve fitting, and the solution of $d_i$ adopts the standard least square method and endpoint constraints. Interpolate the first and last data points $P_0 = p(0)$, $P_m = p(1)$, and other data points $P_i$ ($i = 1, 2, \cdots, m - 1$) are approximated by the least-squares minimization method.

The given data points are parameterized by the normalized accumulation chord length, as shown in Formula (4),

$$x_0 = x_0^* = 0$$
$$x_i^* = x_{i-1}^* + |\Delta P_{i-1}| \quad i = 1, 2, \cdots, m \tag{4}$$
$$x_i = \frac{x_i^*}{x_m^*}$$

where $x_i$ is the parameter value corresponding to the current data point, and $\Delta P_{i-1}$ is the forward difference vector, $\Delta P_{i-1} = P_i - P_{i-1}$.

The objective function is the square difference between each data point and the corresponding point on the curve fitted according to the new knot vectors obtained by HPA in the previous iteration, as shown in Formula (5). Then we can get Formula (7) through the transformation of Formula (6).

$$f = \sum_{i=1}^{m-1} [\, P_i - p(x_i)\, ]^2 \tag{5}$$

Let

$$r_i = P_i - P_0 N_{0,\rho}(x_i) - P_m N_{n,\rho}(x_i) \quad i = 1, 2, \cdots, m - 1 \tag{6}$$

Then,

$$f = \sum_{i=1}^{m-1} [r_i - \sum_{j=1}^{n-1} d_j N_{j,\rho}(x_i)]^2 \tag{7}$$

The $L$-th derivative of it is shown in Formula (8), in order to minimize the objective function $f$, the derivative of $n - 1$ control points $d_j$ needs to be 0, and then the final least-square fitting in Formula (9) can be obtained to solve the control point $d_j$ in the current number of iterations.

$$\frac{\partial f}{\partial d_L} = \sum_{i=1}^{m-1} [-2r_i N_{L,\rho}(x_i) + 2N_{L,\rho}(x_i) \sum_{j=1}^{n-1} d_j N_{j,\rho}(x_i)] \tag{8}$$

$$\sum_{j=1}^{n-1} (\sum_{i=1}^{m-1} N_{L,\rho}(x_i) N_{j,\rho}(x_i)) d_j = \sum_{i=1}^{m-1} r_i N_{L,\rho}(x_i)] \tag{9}$$

According to the obtained basis functions $N_{i,\rho}(x)$ and control points $d_j$, they are brought into Formula (3) to complete the curve fitting.

### 2.2. Research Status

At present, many companies have developed relatively mature ship CAD software systems such as AVEVA's TRIBON Solution, DASSAULT's Catia, NAPA's NAPA system, etc. In addition, In-Il Kim [25], Hyeon-deok Lee [26] and Dongkon Lee et al. [27] also developed a special ship design system. The dedicated ship CAD system effectively improves the efficiency and quality of ship design. However, it has not achieved a better solution to the problem of discrete point processing in the offset table. For example, in the research on the reconstruction of the segmented outer plate of the ship surface, Yu Yong et al. [28] carried out the rapid inverse calculation of the non-uniform B-spline curve, but the inverse calculation process is to interpolate the discrete points, which cannot avoid errors. Nowadays, researchers prefer B-spline approximation. Moreover, the knot

placement problem is a popular content that has been studied by many scholars in recent years [29]. The existing research methods can be divided into the static knot method and dynamic knot method according to whether the knot can be adjusted after selection. We first introduce the relevant static knot methods. Piegl L and Tiller W [23] first proposed an averaging technique and knot placement technique (KTP) in 1978, and knots are selected to reflect the distribution of data points. On this basis, Piegl L and Tiller W put forward NKTP technology [30]. Lyche T proposed a knot removal method (KRM) in 1988 [31,32]. However, this method cannot capture the internal characteristics of data points, and the amount of calculation is huge. In 1999, Razdan A [33] proposed to use the shape information of data points for knot placement. In the research method of Park H and Lee J [34], the idea of knot adaptation was proposed. Xu Jin, Ke Yinglin et al. [35] further proposed a feature point including the crease point, inflection point and curvature maximum point. The influence of curvature on fitting has also become a more concerned factor [36–39]. The static knot method is usually simple to calculate and has higher efficiency; the knot calculation comes from artificial assumptions and cannot be moved after placement. Compared with the dynamic knot method, the static knot method requires more knots under the same precision requirement. The dynamic knot method is mainly a combination of B-spline curve fitting theory and optimization algorithm, and the overall algorithm performance largely depends on the optimization algorithm. There are hundreds of existing optimization algorithms, and the nature-inspired optimization algorithm [40–47] is one of the most practical branches. According to the algorithm principle, it can be divided into three categories, as shown in Figure 1. Yoshimoto F et al. [17] and Sarfraz M [18] proposed the application of genetic algorithm. Özkan İNİK et al. [19] combined the grey wolf algorithm. He Bingpeng et al. [20] combined the differential evolution algorithm. Kübra Uyar et al. [21] combined the invasive weed optimization (IWO). Akemi Gálvez et al. [22] proposed the combination of immune algorithm and B-spline curve fitting and improved the parameter optimization and algorithm complexity. The dynamic knot method realizes the adaptive adjustment of knot dimension and position in each iteration; it does not rely on artificial assumptions and carries out forward calculation through multiple groups of initial random knots. The fitting effect of dynamic knot method is often better than the static knot method, but its operation efficiency is low.

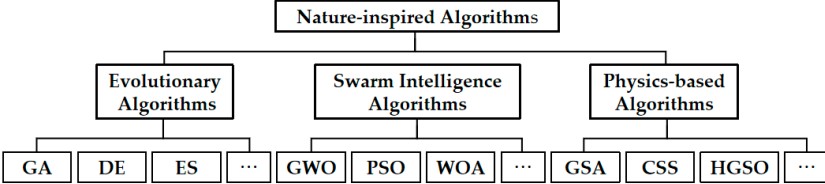

**Figure 1.** Classification of nature-inspired optimization algorithms.

## 3. A B-Spline Curve Fitting of Hungry Predation Optimization Algorithm with Knot Guidance

### 3.1. Knot Guidance Technology

In B-spline curve fitting, zero inner knots represent a straight line. As the shape complexity increases, so does the inner knot number. The knot guidance technology proposed in this paper mainly captures the feature points of the curve through the preprocessing of the curve data points. The feature points include jump points (position discontinuity points), sharp points (tangent discontinuity points), curvature discontinuity points, curvature extreme points and inflection points. This paper mainly focuses on the engineering application of ship line, which includes three types of feature points, as shown in the Figure 2.

According to the Formula (10) [38],

$$K_i = \frac{2\Delta P_{i-1}P_iP_{i+1}}{l_i l_{i+1} l_i^*} = \text{sgn}(\Delta P_{i-1}P_iP_{i+1})\frac{\text{sinff}_i}{l_i^*} \ (i = 1, 2, \dots, m-1) \tag{10}$$

where $K$ is the curvature of the data point; $P$ is the data point; subscript $i$ is the serial number of the corresponding data point; $\Delta P_{i-1}P_i P_{i+1}$ is the triangular area composed of data point $P_{i-1}$, $P_i$ and $P_{i+1}$; sgn is the symbolic function; $\Delta P_{i-1}P_i P_{i+1}$ is positive when the order is counterclockwise; $l$ is the chord length between two points; and $\alpha_i$ is the angle of the chord, as shown in the Figure 3.

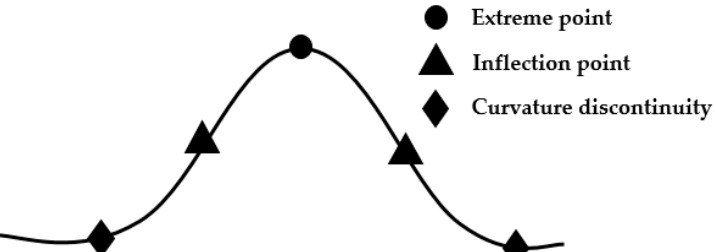

**Figure 2.** Feature points of curve.

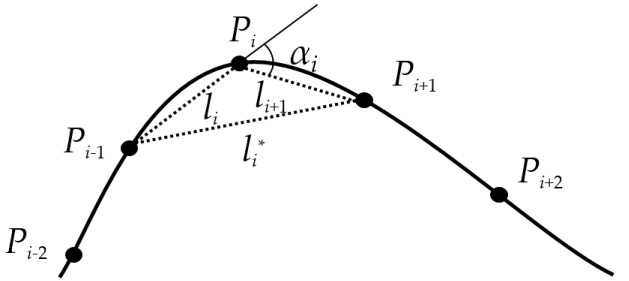

**Figure 3.** Schematic diagram of Formula (9).

After obtaining the curvature of each point, find the position of the feature point and the corresponding parameter value $u_i$. In order to ensure the diversity of the population size, half of population still take random values in the interval [0, 1]. The other half takes values in each interval $[u_i - 1, u_i + 0.1]$; each knot has a 50% chance to mutate, and the value interval after mutation is [0, 1]. The method of finding feature points is described in reference [35].

### 3.2. Hungry Predation Optimization Algorithm

The adaptive adjustment of knots in HPA simulates the predator–prey behavior of animals under starvation. Distinguish individual levels in the population based on fitness function. Those with higher fitness are $\alpha$, and the remaining individuals are collectively referred to as $\omega$.

The three groups of knot vectors with the best fitness and their coordinates are:

$$U_p^{\alpha_1} = (u_p^1, u_p^2, \cdots, u_p^{D\alpha_1})$$

$$U_p^{\alpha_2} = (u_p^1, u_p^2, \cdots, u_p^{D\alpha_2})$$

$$U_p^{\alpha_3} = (u_p^1, u_p^2, \cdots, u_p^{D\alpha_3})$$

Traverse the population in turn, and the corresponding position coordinate is:

$$U_i = (u_i^1, u_i^2, \cdots, u_i^D)$$

If the individual is $\alpha$, then its position remains unchanged. If the individual is $\omega$, the next position $U_{i+1}$ is described as Formulas (11)–(13),

$$a = 2 - Iter/Max\_iter \tag{11}$$

$$U_{i+1}^{\alpha_k} = U_p^{\alpha_k} - A|CU_p^{\alpha_k} - U_i| \; k = 1, 2, 3 \tag{12}$$

$$U_{i+1} = \sum_{k=1}^{3} U_{i+1}^{\alpha_k}/3 \tag{13}$$

where $A$ is a random number between $[-a, a]$, $C$ is a random number between $[0, 2]$ and *Iter* is the current number of iterations.

At this time, the initial adjustment of the population position has been achieved, and each individual has a 50% chance to become a hungry individual. The hungry individual will further hunger search and move faster to the prey position.

First, use the Euclidean distance between the current position $U_i$ and the position $U_\alpha$ of the $\alpha_1$ to calculate the radius $R_i$ of the neighbor range, as shown in Formula (14).

$$R_i = ||U_i - U_{\alpha_1}|| \tag{14}$$

Then, traverse the population size to find neighbors, as shown in Formula (15).

$$N_i = \{U_j | D_i(U_i, U_j) \le R_i\} \tag{15}$$

where $N_i$ is the set of neighbors and $U_j$ is the location of the neighbor.

Finally, move the position, as shown in Formula (16).

$$U_{i+1} = U_r + (U_r - U_i) \times S \times \cos(\frac{Iter}{Max\_iter} - 0.5) \tag{16}$$

where $U_r$ has a 50% probability of being the location of $\alpha$, and 50% is the neighbor in the population size. $S$ is a random number between $(0, 1)$.

*3.3. Fitness Function Selection*

As a key component of the algorithm, the selection of fitness function directly affects whether it can find the optimal solution and the convergence speed of HPA. In the whole process of population size search, it does not rely on external information, only based on the fitness function. Individuals in a population adjust their positions based on fitness.

This paper proposes three fitness functions that can be applied to HPA.

(a) Bayesian Information Criterion (BIC) [22]: its fitness function is described as Formula (17).

$$Fitness = -2\text{Ln}(L_1) + \text{Ln}(n_1) \times N_p \tag{17}$$

where $L_1$ is the likelihood function, $n_1$ is the sample size and $N_p$ is the number of parameters.

(b) Maximum error of a single point of the curve is described as Formula (18).

$$Fitness = Max\left(P_{ir} - P_{if}\right) \tag{18}$$

where $P_{ir}$ is the actual given data point, and $P_{if}$ is the point on the corresponding fitting curve.

(c) Overall standard deviation of curve is described as Formula (19).

$$Fitness = \sqrt{\frac{\sum_{i=1}^{n}\left(P_{ir} - P_{if}\right)^2}{N_k}} \tag{19}$$

where $N_k$ is the number of knot vertices.

The BIC method considers the complexity of the calculation model and avoids the over fitting problem by adding the penalty term of model complexity. When the number of knots and control points is small, the BIC value will decrease with increasing the number of knots until it reaches the minimum value. Then, as the number of knots continues to increase, the accuracy of the curve improves slowly, and the BIC value will increase. In practical engineering application, Formulas (18) and (19) are more intuitive and convenient.

### 3.4. Dimension Calculation Rule

In the iterative calculation of population position, different individuals have different internal knot rates $\lambda$, which causes the dimensions to be different for each individual, as shown in Figure 4. The dimension corresponds to the number of internal knots. In order to find fewer control points to complete the curve fitting, we must ensure the diversity of individuals. When $\omega_i$ moves towards or away from $\alpha$, its dimension remains unchanged. We deal with $\alpha$ whose dimension is different from $\omega_i$. Then we can get $U_p^{\alpha_1}$, $U_p^{\alpha_2}$ and $U_p^{\alpha_3}$, and all of them have the same dimensions with $\omega_i$.

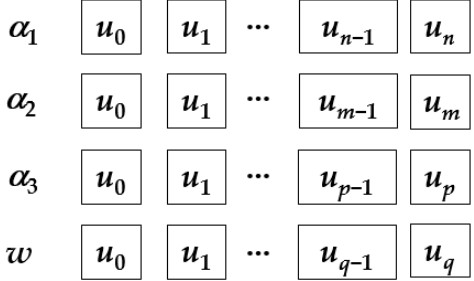

**Figure 4.** Dimensions of different individuals.

First, the dimensions of $\alpha$ are judged. If the dimension is higher than that of the currently selected $\omega_i$, calculate the extra number of dimension $N1$ and the difference $M_i$ between every two adjacent inner knots of the high-dimensional individual.

The probability $p_i$ of each point to be deleted is shown in Formula (20):

$$p_i = \frac{M_i}{\sum_{i=0}^{m} M_i}(i = 0, 1, \cdots, m) \tag{20}$$

Randomly delete a knot according to the probability $p_i$, and this process is repeated $N1$ times until two individuals have the same dimension.

If the dimension is lower than that of the currently selected individual, calculate the extra number of dimension $N1$ and the difference $M_i$ between every two adjacent inner knots of the high-dimensional individual. Randomly add a new knot in the interval with the largest inner knot difference $M_i$ and repeat this process to $N1$ times, so that the dimensions of the two individuals are the same.

### 3.5. Algorithm Flow

The overall algorithm flow of HPA is shown in Figure 5.

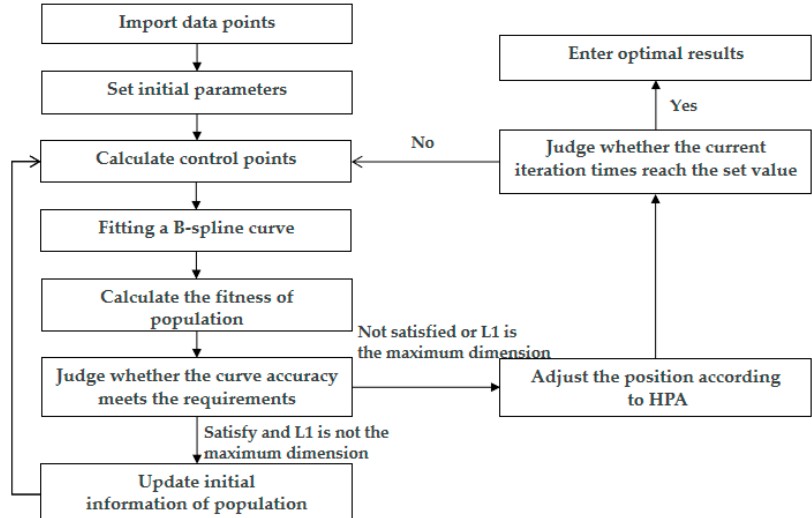

**Figure 5.** Algorithm flow of HPA.

Step 1: Import the data points of the curve, order the data points and carry out the parameterization of the standard accumulation chord length to obtain the corresponding parameter values.

Step 2: Set the error threshold, population size and iteration number, which are adjusted adaptively according to the feature points and error threshold, which is introduced in Section 4. The corresponding internal knots are obtained through the knot guidance technology.

Step 3: Based on the existing knot vector, curve degrees and data point, the respective control points are obtained by using the least-squares minimization method, in which the curve degrees and data point parameters are fixed values, and the value and dimension of each individual's knot vectors may change after each iteration.

Step 4: According to the existing knot vectors and the control points obtained in Step 3, a B-spline curve can be fitted by using Formula (3).

Step 5: Give a fitness function, calculate the fitness of each individual.

Step 6: Judge whether the curve fitted by each individual meets *E*. If the individual meets *E*, and its dimension is not the largest among the population, then go to step 7. If it does not meet *E* or the individual's dimension is the largest among the population, go to step 8.

Step 7: Determine the minimum dimension *D* of population that satisfies the condition, remove individuals whose dimension is greater than *D*, supplement the population with the same number of individuals with dimension of *D* or *D*−1 and return to step 3.

Step 8: Use the hungry predation optimization technology to adjust the position of each individual.

Step 9: Judge whether the current iteration number reaches the set value *Max_iter*. If it reaches, output the current optimal result. If it does not reach *Max_iter*, return to step 3.

## 4. Key Parameter Settings

There are four key parameters of the HPA:

(1) Population size *W*, which determines how many individuals find the optimal solution in the population size at the same time.

(2) Internal knot rate λ: The range of λ depends on the number of given data points, usually select interval [0, 0.5], which can be reduced with the increase of data points. The λ is different for each individual.

(3) Error threshold *E*, which ensures that the error value of the final fitted curve is less than *E*.

(4) Iteration number *Max_iter*, which determines the number of iterations of the whole optimization process, iterates to the *Max_iter* time and outputs the current optimal solution.

In previous studies [17–22], the population size and iteration number have been based on different fitting models and fitting error thresholds, relying on intuitive experience or multiple attempts to give appropriate values, which is troublesome and time consuming. This section takes two continuous semicircle models with radius 1 as an example, as shown in Figure 6. We further explore the impact of population size, iteration number and error threshold on curve fitting effect, and give the setting formula of population size and iteration number.

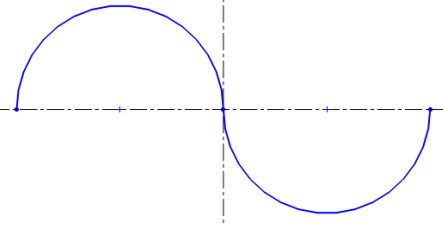

**Figure 6.** Two continuous semicircles. Radius: 1.

The population size determines the number of parallel solution paths. When the iteration number *Max_iter* is 20 and the error threshold *E* is 0.1, the population size *W* is set to 10, 20 and 50, respectively; the fitting result is recorded as shown in Figure 7.

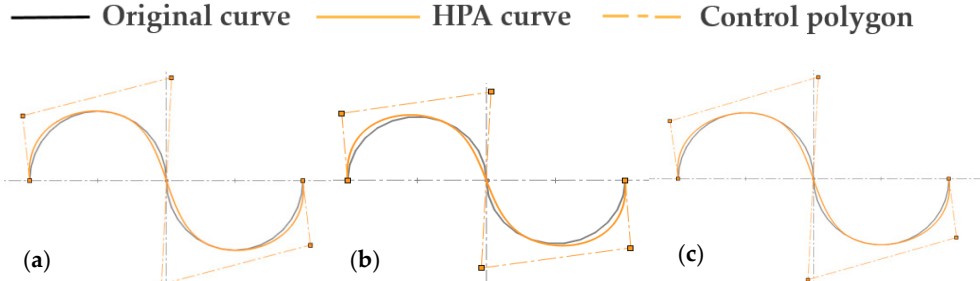

**Figure 7.** Influence of population size *W* on curve fitting, (**a**) *W* = 10, (**b**) *W* = 20 and (**c**) *W* = 50.

The inner knots of Figure 7a are 0.4328 and 0.5771, and the fitting accuracy is 0.0186. The inner knots of Figure 7b are 0.4612 and 0.5376, and the fitting accuracy is 0.0207. The inner knots of Figure 7c are 0.4076 and 0.5924, and the fitting accuracy is 0.0169. It can be seen from Figure 7 that the fitting result with *W* = 20 is worse than *W* = 10 in this experiment.

The setting of the iteration number affects the step size of each iteration. The more the iteration number, the more carefully the algorithm searches. However, when the iteration number is large enough, increasing the iteration number will have little effect on the curve fitting accuracy. Besides, a large iteration number will improve the operation time of the algorithm. When the error threshold *E* is 0.1, multiple experiments are carried out by changing the population size *W* and iteration number *Max_iter*. The results are shown in Figure 8.

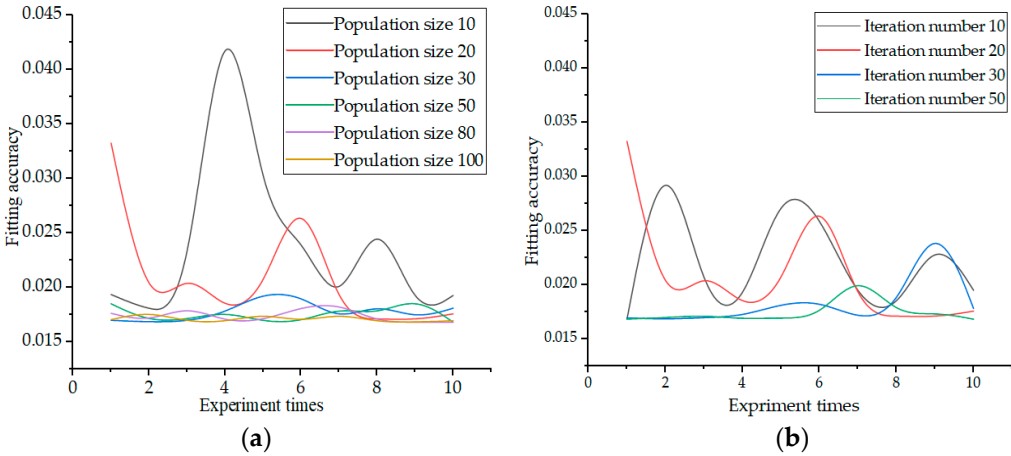

**Figure 8.** Fluctuation of curve fitting accuracy when (**a**) *Max_iter* is 20 and *W* varies from 10 to 100, and when (**b**) *W* is 20 and *Max_iter* varies from 10 to 50.

As shown in Figure 8a, *Max_iter* is 20. When *W* is 10 and 20, the result fluctuates greatly, and the fluctuation is relatively small when *W* is 30. In Figure 8b, *W* is 20, and the fluctuation is small after 30 iterations.

The setting of the error threshold determines the fitting accuracy of the B-spline curve. The HPA algorithm can better implement the B-spline curve with the least number of control points when the error threshold is satisfied. Set *W* to 30, *Max_iter* to 20 and set *E* to 0.1, 0.01 and 0.001, respectively; the fitting results are shown in Figure 9.

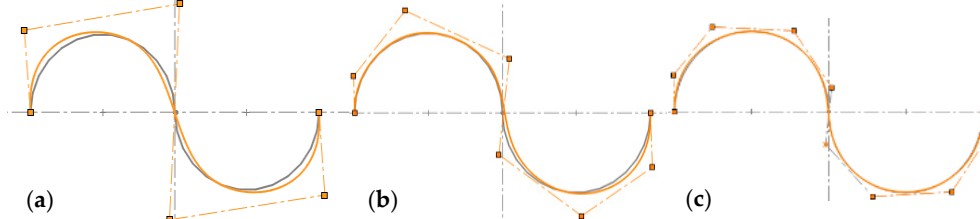

**Figure 9.** When *E* is (**a**) 0.1, (**b**) 0.01 and (**c**) 0.001, the influence of their changes on curve fitting.

The least-squares fitting accuracies of Figure 9a–c are 0.0207, 0.0016, and 0.0004, respectively. It can be seen from the results that as *E* decreases, the final curve fitting result is getting better, and more control points are required.

Adjust *E* from 1 to 0.0001, and the change of inner knot number required for the fitted curve, which is shown in Figure 10.

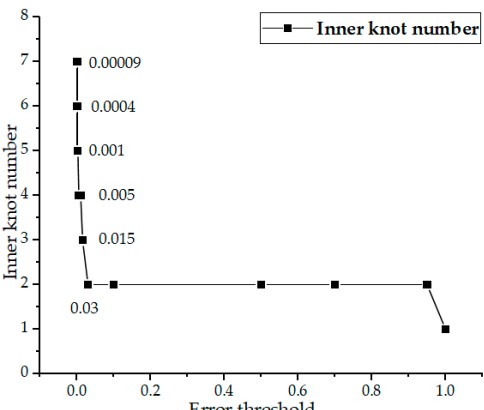

**Figure 10.** Influence of error threshold on the inner knot number.

The optimization algorithm in the field of B-spline curve fitting is to solve the extremum problem, and the solution is a number of interrelated decimals arranged in ascending order from 0 to 1. The setting of population size and iteration number is mainly related to the number and distribution of solutions. It is affected by factors such as error threshold and model complexity. A small numerical setting will make it difficult to find the optimal solution, and a large numerical setting will lead to the algorithm low efficiency. Therefore, before the algorithm runs, we need to set a reasonable population size *W* and iteration number *Max_iter*. This paper gives the relational Formulas (21) and (22).

$$W = \left\lfloor -\text{In}\left(\frac{E}{L_{MAX}}\right) \times 1.2^{\text{Log}_{10}\left(\frac{L_{MAX}}{E}\right)} + 1.5 \times Cp \right\rfloor \tag{21}$$

$$Max\_iter = 2 \times W \tag{22}$$

where $L_{MAX}$ is the maximum side length of the model, $Cp$ is the number of model feature points and $\lfloor\ \rfloor$ is rounded down.

The design of Formulas (21) and (22) makes the HPA algorithm only need to input the model data points and the required error threshold, and the HPA algorithm can adaptively adjust the population size and iteration number. The whole algorithm has better universality, and it is ensured that the algorithm can obtain a better fitting effect at a higher operating efficiency.

## 5. HPA Algorithm Test

The experiment is divided into two parts. Section 5.1 compares the static knot method and dynamic knot method, and Section 5.2 compares with the existing commercial Software Solidworks2022 (Dassault, US) and Catia2017 (Dassault, France).

### 5.1. Comparison with Existing Research

When we deal with the problem of knot placement, except for the method of combining optimization algorithm, B-spline curve fitting based on adaptive curve refinement using dominant points (DOM), KTP, NKTP, KRM etc. are the best representative methods, as shown in Table 1. Figure 11 is a comparison example of HPA in this paper and DOM method. Park H compared related algorithms, as shown in Figure 12. Furthermore, we add the fitting results of GWO and HPA to Figure 12. Compared with related methods, HPA uses less control points under the same accuracy.

**Table 1.** Comparison of related fitting methods.

| | Incremental Method Using KTP or NKTP | Knot Removal Method KRM | Incremental Method Using DOM | Adaptive Adjustment Using HPA |
| --- | --- | --- | --- | --- |
| Preprocessing | 1 Parameterization | 1 Parameterization<br>2 Interpolation of all points | 1 Parameterization<br>2 Selection of seed points | 1 Parameterization<br>2 Find feature points |
| Iteration process | 1 Knot placement<br>2 Least-squares minimization<br>3 Deviation check | 1 Selection of a candidate knot<br>2 Deviation check<br>3 Knot removal | 1 Knot placement<br>2 Least-squares minimization<br>3 Deviation check<br>4 Selection of a new dominant point | 1 Knot placement<br>2 Least-squares minimization<br>3 Comparative fitness<br>4 Knot adaptive adjustment |
| Ref. | [23,30] | [31] | [34] | Our method |

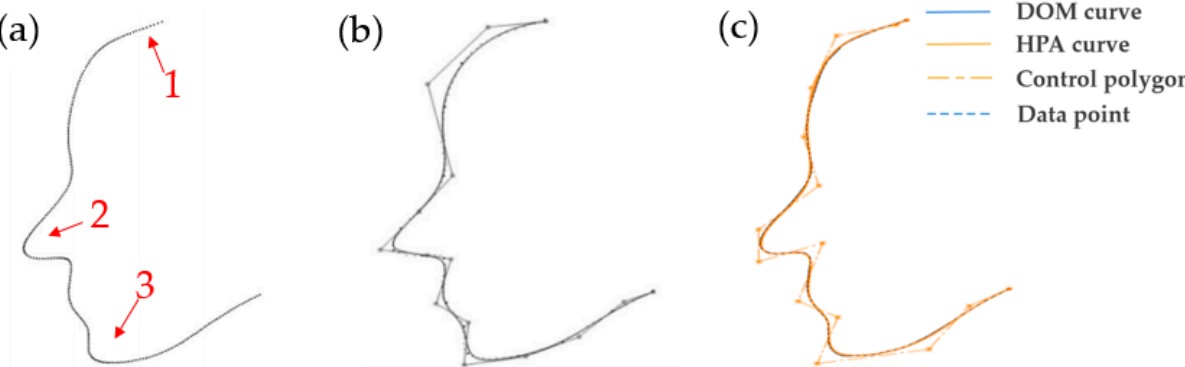

**Figure 11.** Comparing (**a**) the face data points and the fitting results of (**b**) DOM and (**c**) HPA especially at 1–3 marked.

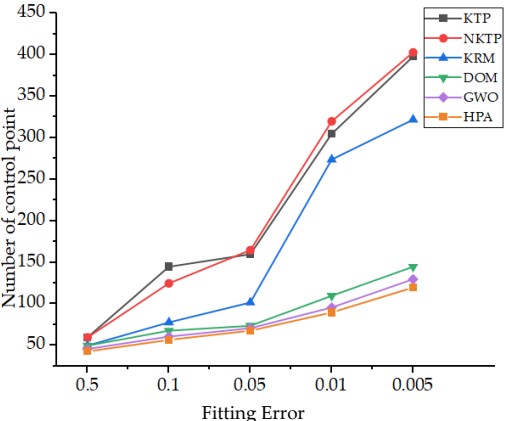

**Figure 12.** Comparison of control point number used at the same fitting error with DOM, KTP, NKTP, KRM, GWO and HPA.

As shown in Figure 11, the same face contour is fitted with the same 15 control points. From the fitting results, it can be seen that the HPA proposed in this paper can better

represent the curve shape of face contour, especially at specially marked 1, 2 and 3 points, which are fuller at the tip of nose and more prominent at the chin compared with DOM.

Operational efficiency is an important criterion for judging the pros and cons of an algorithm. However, due to the different research periods of each algorithm, there are differences in both software and hardware. It is not objective to simply compare the running time with previous literature. The optimization algorithms of the natural heuristic class take a similar time for each iteration, and it is fairer to compare the convergence of the same number of iterations. This paper selects six representative optimization algorithms for face contour fitting: Genetic Algorithm [40], Differential Evolution Algorithm [41], Grey Wolf Algorithm [42,43], Whale Algorithm [44], Particle Swarm Algorithm [45] and Gravity Algorithm [46]. The search ability and convergence efficiency of HPA and other algorithms are compared when using 15 control points, as shown in Figure 13.

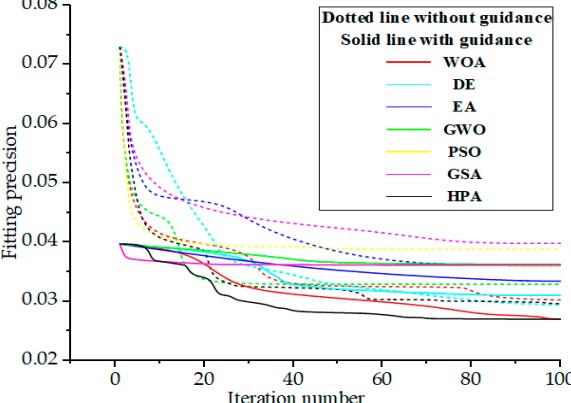

**Figure 13.** Comparison of the effect of various optimization algorithms on face contour fitting.

It can be seen from the above experiments that after knot guidance technology, the algorithm can find a better solution in the first generation. Compared with other algorithms, HPA tends to have higher fitting accuracy and achieve faster convergence at the same number of iterations. Besides, DE, WOA and HPA demonstrate stronger search capabilities. The DE algorithm uses a greedy algorithm, which makes the entire calculation time approach twice that of other algorithms.

The paper further selects the six models mentioned in the literature [19] for testing, as shown in Table 2, and compares them with the GWO method mentioned in the literature. The results of fitting the model are shown in Figure 14 and Table 3.

**Table 2.** Test six models mentioned in the literature. (Adapted with permission from Ref. [19]).

| Fuction | Description | Variable Range |
|---|---|---|
| 1 | Data point [48] | |
| 2 | $f_2(x) = \frac{10x}{(1+100x^2)}$ | $[-2, 2]$ |
| 3 | $f_3(x) = 0.2e^{-0.5x}\sin 5x + 4$ | $[0, 4\pi]$ |
| 4 | $f_4(x) = \frac{100}{e^{|10x-5|}} + \frac{(10x-5)^5}{500}$ | $[0, 1]$ |
| 5 | $f_5(x) = \sin x + 2e^{-30x^2}$ | $[-2, 2]$ |
| 6 | $f_6(x) = \sin 2x + 2e^{-36x^2} + 2$ | $[-2, 2]$ |

It can be seen that the fitting results of the HPA method proposed in this paper for the six models are much better than GWO method at the same population size, iteration number, data points and number of knots.

The test function $f_4(x)$ is a more challenging function in the field of B-spline fitting [22], and there is a sharp point at $x = 0.5$, which is a commonly used example in the B-spline curve fitting literature. Existing fitting methods based on optimization algorithm and their

fitting results are shown in the Table 4; PESA, MOGA, FFA and some other methods cannot fit the $f_4(x)$ function well. BIC is a criterion for many researchers to judge the pros and cons of an algorithm. However, each scholar has a slightly different understanding of the parameters in the BIC formula. Therefore, the BIC judgment formula in the B-spline curve fitting application is different.

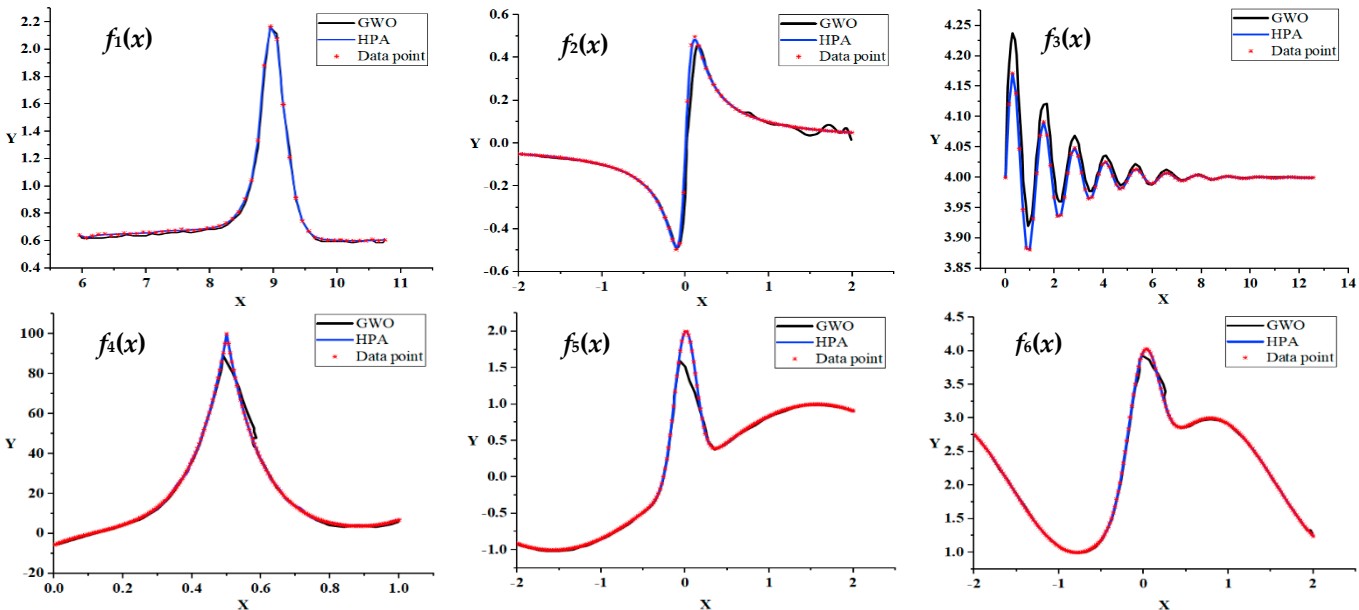

**Figure 14.** Comparison of GWO and HPA fitting results for six test models.

**Table 3.** The experimental results for all functions.

| Fuction | Degree of Curve | Population Size | Number of Iteration | Number of Point | Number of Knot | Number of Control Point | MSE (GWO) | MSE (HPA) |
|---|---|---|---|---|---|---|---|---|
| 1 | 3 | 50 | 100 | 49 | 16 | 20 | 0.024 | $3.21 \times 10^{-3}$ |
| 2 | 3 | 50 | 100 | 90 | 53 | 57 | 0.010 | $6.32 \times 10^{-7}$ |
| 3 | 3 | 50 | 100 | 200 | 77 | 81 | 0.008 | $2.31 \times 10^{-6}$ |
| 4 | 3 | 50 | 100 | 201 | 40 | 44 | 1.395 | $3.18 \times 10^{-3}$ |
| 5 | 3 | 50 | 100 | 201 | 46 | 50 | 0.032 | $2.43 \times 10^{-4}$ |
| 6 | 3 | 50 | 100 | 201 | 37 | 41 | 0.026 | $7.58 \times 10^{-4}$ |

**Table 4.** Existing fitting methods based on optimization algorithm and their fitting results. ('×' means that the method can not fit the function $f_4$ well).

| Authors, Year and Reference | Method | Iteration Number | Knot Number | BIC |
|---|---|---|---|---|
| Yoshimoto et al. [17] | Genetic algorithms (GA) | 200 | 5 | 1188 |
| Sarfraz and Raza [18] | GA and Detection Algorithms | 120 | × | × |
| Özkan İNİK et al. [19] | Gray Wolf Optimization (GWO) | 100 | 40 | 704 |
| Kübra Uyar et al. [21] | Invasive Weed Optimization (IWO) | 15 | 6 | 430 |
| Gálvez et al. [22] | Artifficial Immune Systems (AIS) | 100 | 5 | 1121 |
| Ulker and Arslan [49] | Artificial Immune Systems | 500 | × | × |
| Ulker [50] | Pareto Envelope-Based Selection Algorithm (PESA) | 500 | × | × |
| Valenzula et al. [51] | Multi-Objective Genetic Algorithms (MOGA) | 120 | × | × |
| Gálvez and Iglesias [52] | Fireflfly Algorithm (FFA) | 500 | × | × |
| Yuan et al. [53] | Adaptive Multiresolution Basis Set with Lasso Selection Method | Variable | × | × |

In the AIS [22] method, the BIC calculation method is shown in Formula (23).

$$BIC = N \times LN(Q_1) + (LN(N))(2Nod - 3\rho^* + 2)$$
$$Q_1 = \sum_{i=1}^{N} (X_{1i} - X_{2i})^2 + (Y_{1i} - Y_{2i})^2 \tag{23}$$

where $N$ is the number of data points, *Nod* is the number of knots and $\rho^*$ is the order of the curve; $X_{1i}$ and $X_{2i}$ are the X coordinate values of the *i*-th point on the actual curve and the fitted curve, respectively; $Y_{1i}$ and $Y_{2i}$ are the Y coordinate values of the *i*-th point on the actual curve and the fitted curve, respectively.

In the testing process of the AIS method, 201 data points are selected at equal distances. The degree of curve $\rho$ is 3 ($\rho^* = 4$). In order to further test the ability of the algorithm to withstand interference, a normal fluctuation with a mean of 0 and a variance of 1 was applied to the data points. The minimum BIC value is 1121.09 when $k = 5$. Under the same BIC calculation formula, the HPA algorithm obtains the smallest BIC value of 1040.57 when $k = 9$, as shown in Table 5.

**Table 5.** BIC values of AIS and HPA methods at the same knot.

| Inner Knot Number | Method | $Q_1$ | RMSE | BIC |
|---|---|---|---|---|
| $\kappa = 3$ | HPA | 500.889 | 1.578 | 1313.12 |
| | AIS | 593.977 | 1.719 | 1336.79 |
| $\kappa = 4$ | HPA | 329.117 | 1.279 | 1239.32 |
| | AIS | 381.871 | 1.378 | 1258.60 |
| $\kappa = 5$ | HPA | 169.758 | 0.919 | 1116.86 |
| | AIS | 182.761 | 0.953 | 1121.09 |
| $\kappa = 6$ | HPA | 142.331 | 0.842 | 1092.05 |
| | AIS | 181.051 | 0.949 | 1129.81 |
| $\kappa = 7$ | HPA | 122.786 | 0.782 | 1072.96 |
| | AIS | 178.013 | 0.941 | 1137.01 |
| $\kappa = 8$ | HPA | 102.723 | 0.715 | 1047.71 |
| | AIS | 176.391 | 0.936 | 1145.78 |
| $\kappa = 9$ | HPA | 94.045 | 0.684 | 1040.57 |
| | AIS | 174.515 | 0.931 | 1154.24 |
| $\kappa = 10$ | HPA | 92.984 | 0.680 | 1043.60 |
| | AIS | 172.218 | 0.919 | 1159.61 |

In the Gray Wolf Optimizer (GWO) for knot placement in B-spline curve fitting [19], the BIC calculation formula is shown in Formula (24). The HPA proposed in this paper is much better than GWO in fitting the function $f_4(x)$.

$$\text{BIC} = N \times \text{LN(MSE)} + (\text{LN}(N))2 \times (2 \times Nod + \rho^*)$$

$$\text{MSE} = \frac{1}{N} \sum_{i=1}^{N} \sqrt{(X_{1i} - X_{2i})^2 + (Y_{1i} - Y_{2i})^2} \tag{24}$$

where $N$ is the number of data points, *Nod* is the number of knots and $\rho^*$ is the order of the curve; $X_{1i}$ and $X_{2i}$ are the X coordinate values of the *i*-th point on the actual curve and the fitted curve, respectively; $Y_{1i}$ and $Y_{2i}$ are the Y coordinate values of the *i*-th point on the actual curve and the fitted curve, respectively.

The design of the BIC value in this article refers to the mean square error. Compared with the variance, the value will be smaller under the same fitting situation and may even be negative. When the optimal solution fitted by the GWO method is 40 knots, the BIC value is 704. However, the HPA algorithm obtains a minimum BIC value of 230.31 at $k = 4$, as shown in Table 6. Figure 15 records the fitting results of different numbers of knots of test function $f_4(x)$.

When the number of knots is small, the increase of knots can greatly improve the curve fitting accuracy, and the BIC value will decrease accordingly. When the inner knot is 4, the shape of the curve can be better fitted. At this time, the BIC value reaches the minimum value. As the number of knots continues to increase, the improvement of fitting accuracy begins to decrease, and the BIC value begins to increase.

In the GWO method, when 40 knots are used, the minimum BIC value [19] is obtained, and the fitting result of the test function $f_4(x)$ shows that the final fitting effect is not ideal, as shown in Figure 14. It can be seen that the HPA algorithm proposed in this paper achieved an accurate fitting of data points when using 40 knots, while the GWO method still cannot solve the cusp fitting when there are 40 knots. We guess that the use of heavy knots may be ignored in GWO.

**Table 6.** BIC value and knot distribution of HPA with different knot number.

| Inner Knot Number | MSE | BIC | Knot Distribution |
|---|---|---|---|
| κ = 2 | 2.737 | 446.37 | 0.311, 0.372 |
| κ = 3 | 1.015 | 268.32 | 0.282, 0.528, 0.604 |
| κ = 4 | 0.756 | 230.31 | 0.192, 0.505, 0.505, 0.505 |
| κ = 5 | 0.735 | 245.76 | 0.152, 0.472, 0.472, 0.472, 0.717 |
| κ = 6 | 0.707 | 259.12 | 0.151, 0.476, 0.476, 0.476, 0.697, 0.819 |
| κ = 7 | 0.643 | 261.50 | 0.230, 0.309, 0.470, 0.470, 0.530, 0.673, 0.785 |

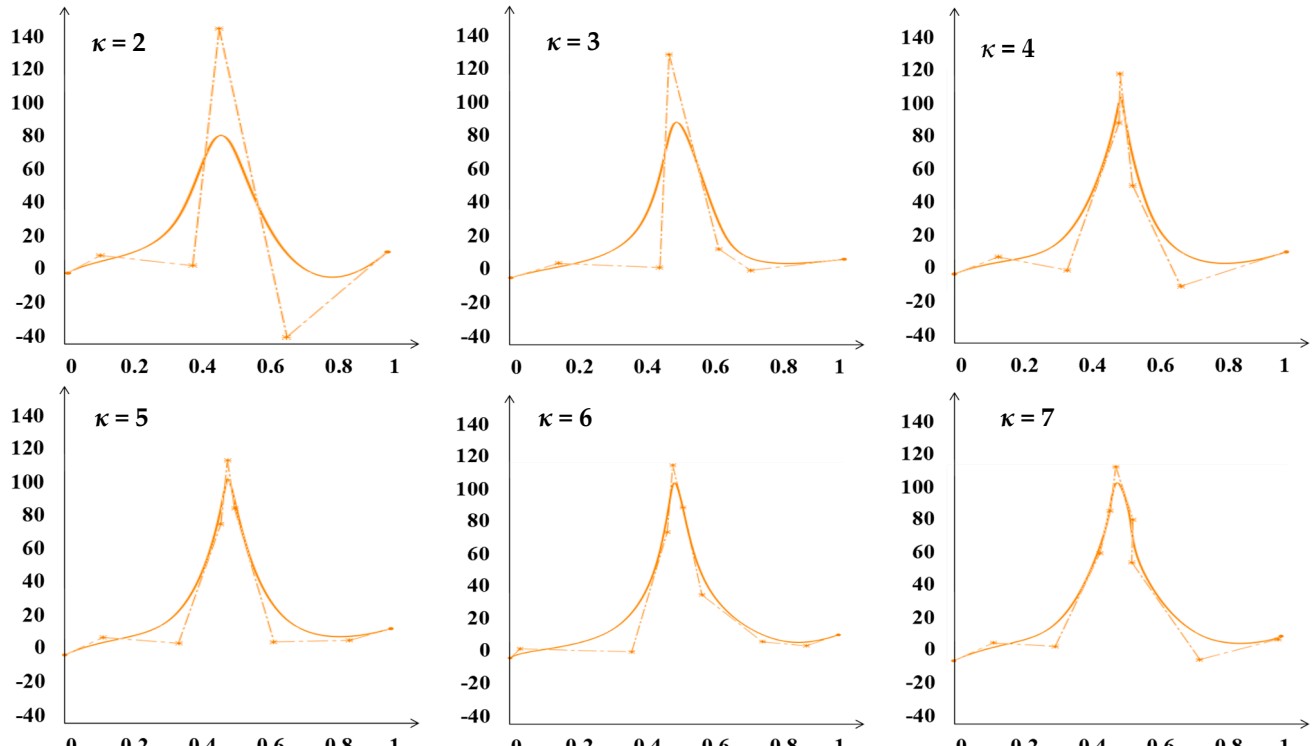

**Figure 15.** Fitting results of different number of knots.

## 5.2. Fitting of Ship Line

Figure 16 shows the offset table and ship line diagram of a ship. The ship has a total length of 189.98 m, a shape width of 32.26 m and a shape depth of 16 m.

Solidworks and Catia are popular design softwares at present. In particular, Catia has more prominent capabilities in complex surface design and reverse engineering. They are widely used in shipbuilding, aviation and other industries. Take the complex ship line on the cross section, longitudinal section and waterline surface and fit it through the existing design software Solidworks, Catia and HPA. Figure 17 compares the maximum error of Solidworks, Catia and HPA fitting curves under the condition of the same number of control points.

**(a)** Offset table

**(b)** Cross section bow and stern

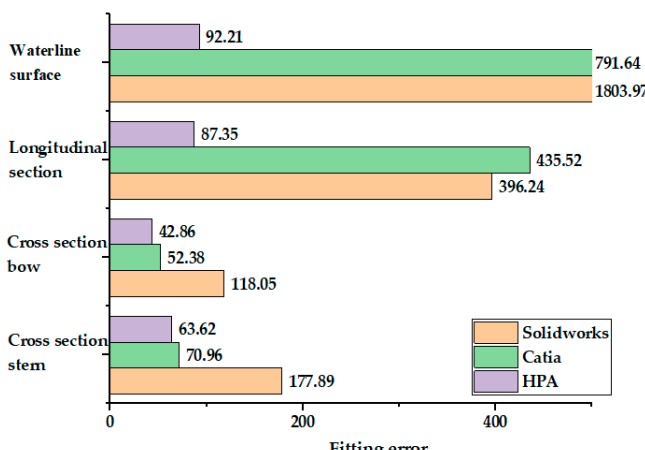

**(c)** Longitudinal section

**(d)** Waterline surface

**Figure 16.** (**a**) Offset table and ship line diagram, which include (**b**) cross section bow, (**c**) longitudinal section and (**d**) waterline surface.

**Figure 17.** Maximum error of fitting using Solidworks, Catia and HPA at the same number of control points.

Input the same data points into Solidworks and Catia and use 10 control points to fit the ship line of the cross section stern, 6 control points to fit the cross section bow, 14 control points to fit the longitudinal section and 12 control points to fit the waterline surface. From the fitting results, it can be seen that under the same number of control points, Catia's fitting accuracy is higher than Solidworks, but both of their overall fitting results are less

effective, especially in the part with continuous curvature change. Given the same data points and the same number of control points, the HPA proposed in this paper has higher precision and can better represent the shape of the curve than Solidworks and Catia.

Figure 18 records the fitting of the ship line of cross section bow, cross section stern, longitudinal section and waterline surface by Solidworks, Catia and HPA. We test the number of control points required by different methods when the fitting accuracy is similar. Figure 19 shows the fitting results with the same number of control points described in Figure 17.

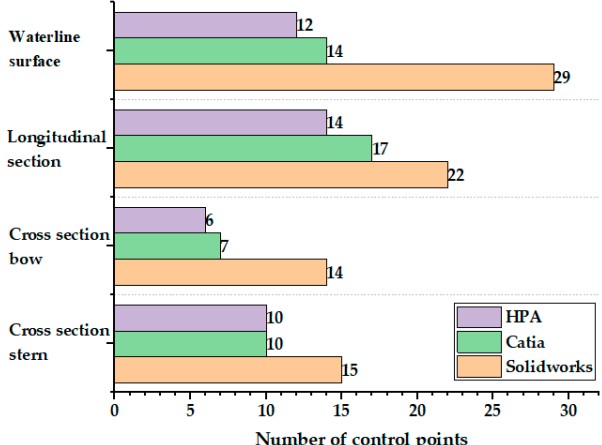

**Figure 18.** Number of control points when the maximum single point error is within 0.1 m.

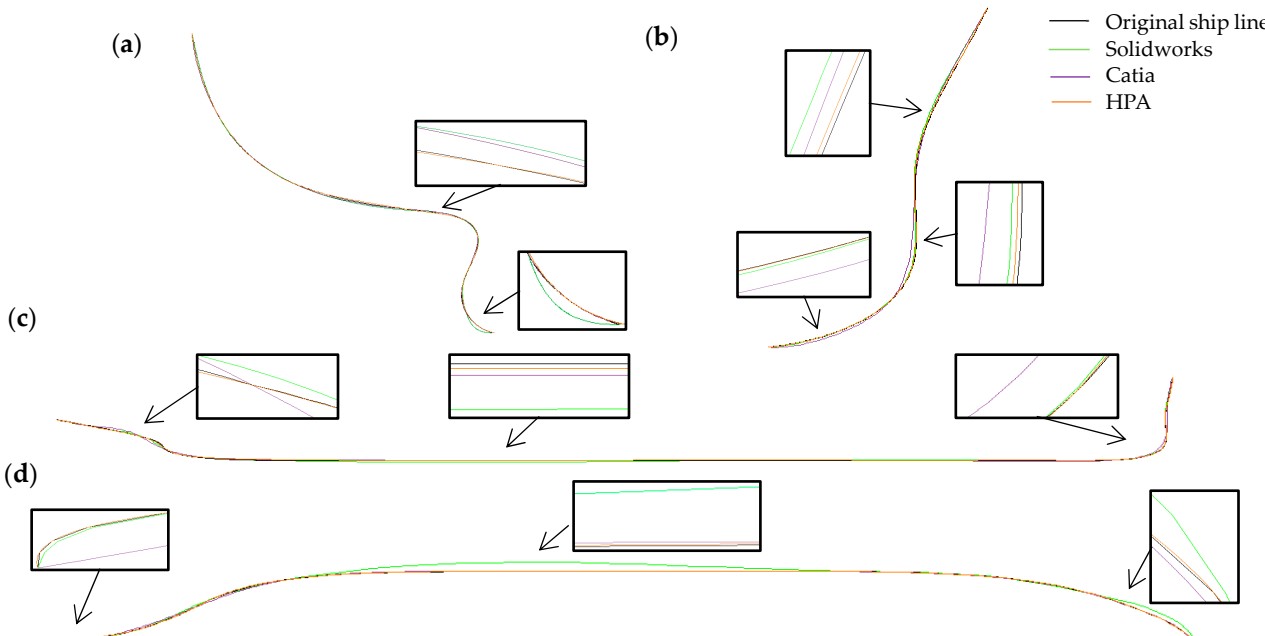

**Figure 19.** Fitting results of ship line of (**a**) cross section stern, (**b**) cross section bow, (**c**) longitudinal section and (**d**) waterline surface with Solidworks, Catia and HPA.

Through the fitting results, it can be seen that under the input of the same data points and the approximate fitting accuracy, compared with the existing design software Solidworks and Catia, HPA can complete the fitting with the same effect with fewer control points, which better proves the superiority of the method proposed in this paper.

## 6. Discussion and Conclusions

This paper develops a B-spline curve fitting of hunger predation optimization on ship line design (HPA). In the previous dynamic knot method, the initial knot is randomly selected from the interval [0, 1], which causes many unreasonable knot vectors. Therefore, a large population size and iteration number are required, which leads to a decrease in the efficiency of the algorithm. The knot guidance technology is designed to add knots in the area with complex model shape at the initial knot selection stage. This aims at the problem that the population loses diversity prematurely in the optimization algorithm, which leads to slow convergence and easy to fall into local optimality. A hunger algorithm search strategy is developed to make the hungry individuals in the population size appear near the optimal solution more quickly, and the influence of the neighbors in the population size on their position adjustment is further considered. This aims at the problem that the previous dynamic knot method requires manual adjustment of key input parameters such as population size and iteration number, which is troublesome and time consuming. An adaptive adjustment of key input parameters in HPA algorithm is proposed, which can quickly adapt to the replacement of model and fitting accuracy. We compared with the typical static knot method, dynamic knot method and the existing commercial Software Solidworks and Catia, and the feasibility and superiority of HPA algorithm, are verified. HPA achieves the goal of B-spline curve fitting with higher fitting accuracy at the same control points, less control points under the same accuracy requirements, faster operation efficiency and better universality. HPA can better solve practical engineering problems.

Future work includes extending HPA to B-spline surface fitting, the application of the hunger predation algorithm in other fields and further improving the fitting accuracy and fitting efficiency of B-spline curve fitting.

**Author Contributions:** Conceptualization, C.S. and M.L.; methodology, C.S. and M.L.; software, M.L.; validation, M.L. and S.G. All authors have read and agreed to the published version of the manuscript.

**Funding:** This research was funded by Unveiling and Commanding Science and Technology Project of Liaoning Province (No.2021JH1/10400093); National Natural Science Foundation of China (51305052); Key Laboratory for Precision & Non-traditional Machining of Ministry of Education of Dalian University of Technology, China (JMTZ202001).

**Institutional Review Board Statement:** Not applicable.

**Informed Consent Statement:** Not applicable.

**Data Availability Statement:** Data will be made available on reasonable request.

**Acknowledgments:** This work was supported by Unveiling and Commanding Science and Technology Project of Liaoning Province (No.2021JH1/10400093); National Natural Science Foundation of China (51305052); Key Laboratory for Precision and Non-traditional Machining of Ministry of Education of Dalian University of Technology, China (JMTZ202001).

**Conflicts of Interest:** The authors declare no conflict of interest, and there is no copyright issue in all of figures.

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
