# Peer review of "B-Spline Curve Fitting of Hungry Predation Optimization on Ship Line Design"

_applsci, doi:10.3390/app12199465_

Round 1

Reviewer 1 Report

The paper matches the Journal's items, proposing a hungry predation B-spline curve fitting algorithm with knot guidance (HPA), which includes knot guidance technology, hungry predation optimization technology and adaptive adjustment of algorithm input parameters.

The paper shows some interesting contents that, however, can be sometimes illustrated in a better way.

Here below authors can find some useful suggestions for a prompt revision of the manuscript prior to publication:

- The title of the paper can be slightly modified in relation to the selected topic which is presented in the abstract, where the results of the work can be better illustrated, too;

- Remove in Abstract "which includes knot guidance technology", it is already explained previously with "with knot guidance";

- The description of the sections inserted at the end of section 2 ("The rest of the paper is organized as follows. Section 3 proposes……………….. Finally, we summarize this paper and further prospect the future research direction.") should be moved to the Introduction section, as a conclusive part that explains the entire structure of the paper, section 2 included;

- Check the numeration of the sections since "Discussion and conclusion" must be changed into 6;

- Some sentences could be better formulated and written differently, as for example in the Introduction section:

 "In addition, the data points in the ship-shape line diagram are discrete, two adjacent discrete points are connected by straight line segments, resulting in poor smoothness, causing trouble to the subsequent ship-shape line setting out and processing, affecting the quality and efficiency of shipbuilding." or in section 2 "For the selection of B-spline degree, in theory, the increase of degree will increase the number of control vertices, and the degree of freedom and flexibility of the curve will become better, which can form more complex curves." or here also "For different shape problems, if high-order curves are used to represent, the corresponding appropriate degree cannot be given directly, so that a large number of calculations and tests are required";

- The authors sometimes should be more specific in the terminology used. For example, in the Introduction section, in the following sentence: "With the development of computer-aided design (CAD) and the rise of advanced manufacturing technology, related technologies are widely used in the design and manufacturing industry of free curves and surfaces [1-5].", when referring to "related technologies" they could be more precise; or in section 4 (However, when the iteration number is greater than a certain value, increasing the iteration number has little effect on the curve fitting accuracy, and will greatly increase the iteration number, which improves the operation time of the program.) or other repetitions like in the conclusion section ("We compared with the typical static knot method, dynamic knot method and the existing commercial Software Solidworks and Catia, the feasibility and superiority of HPA algorithm are verified. We compared with the typical static knot method, dynamic knot method and the existing commercial Software Solidworks and Catia, the feasibility and superiority of HPA algorithm are verified".);

- It is recommended to avoid any repetition of vocabulary (see "Because of the locality of B-spline, it can be locally modified on the existing ship-shape lines, which can make the design of ship-shape line more convenient.");

- Review text formatting and page layout on page 3 and page 18;

- Review formatting of formula (3);

- Review layout of Table1 and Table2;

- Review the legend of Figure 11;

- Review the layout of the graphs in figure 14, fig.15 and fig.16;

- The manuscript should be shortened;

- Finally, pay attention to carefully removing the various errors in the manuscript; for example, check the following sentences and correct.

Abstract

- It is difficult to modify the ship-shape line described by the type value table of discrete structure, which often leads to the problems that three view coordinates may not correspond and the error is difficult to con- trol.

- The effectiveness and superiority of the proposed HPA algorithm are verified by comparing with related algorithms and engineering softwares such as Catia.

Introduction

- B-spline curve approximation is a hot and difficult topic in the field of computer- aided geometric design in recent years.

- With the continuous maturity of the shipbuilding industry, lean production has become the development trend of the current shipbuilding industry.

- Compared with interpolation, B-spline curve approximation can better reflect the curve shape and the smooth-

ness of the final curve is better, but its application is also more difficult.

Section 2

- Compared with interpolation, B-spline curve approximation can better reflect the curve shape for a given data point, and the smoothness of the final curve is better, but its application is also more difficult than interpolation.

- Static knot methods are as follows. Pi

- The static knot method is usually simple to calculate and has higher computational efficiency, but the knot position calculation comes from artificial assumptions and cannot be moved after placement, which causes more control vertices are required to complete the curve fitting, and the fitting accuracy is lower.

Section 3

- There are no jump points and sharp points, focusing on curvature discontinuity points, curvature extreme points and inflection points, as shown in the Fig.2.

- l is the chord length between two points, and ai is the included angle of the chord, as shown in the Fig.3.

- .... half of population still take random values in the interval [0,1], and half of population take equal probability values in each interval [xi-0.1, xi+0.1].

- The method of finding characteristic points is better described in document [28].

- According to the fitness function, individuals' level in the population are distinguished,

- Different fitness function designs make the same individual has different status in the population.

- In practical engineering application, especially in shipbuilding industry, the complexity of ship-shape line is moderate, since formulas (17) and (18) only need to calculate error, it is more intuitive and convenient in use.

- ...... the coordinate dimension of each individual are different accordingly,

- Instead, we process α, which dimensions are different from ωi. A

- ..... so that the two individual have the same dimension, and then iterate the position.

Section 4

- (1) Population size W, i.e. population size which determines how many individuals find the optimal solution in the population size at the same time.

- and every individual's λ are not the same.

- ........ and every individual's λ are not the same.

- The inner knots of Fig.7a are 0.4328 and 0.5771, and the least square fitting accu- racy is 0.0186; The inner knots of Fig.7b are 0.4612 and 0.5376, and the least square fitting accuracy is 0.0207; The inner knots of Fig.7c are 0.4076 and 0.5924, and the least square fitting accuracy is 0.0169.

Section 5

- Also, we add the results of GWO and HPA method proposed in this paper to Fig.12 to compare it with the related methods.

- In terms of search ability, DE, WOA, especially the HPA algorithm, showed more powerful ability.

- The statistics of related research in this paper are shown in the Table 4,

Author Response

Response to Reviewer 1 Comments

Point 1: The title of the paper can be slightly modified in relation to the selected topic which is presented in the abstract, where the results of the work can be better illustrated, too;

Response 1: We have revised the title. The new title is B-spline curve fitting of hunger predation algorithm on ship line design.

Point 2: The description of the sections inserted at the end of section 2 ("The rest of the paper is organized as follows. Section 3 proposes……………….. Finally, we summarize this paper and further prospect the future research direction.") should be moved to the Introduction section, as a conclusive part that explains the entire structure of the paper, section 2 included;

Response 2: We adjusted the structure of the article and put the description of the sections in the introduction.

Point 3: Some sentences could be better formulated and written differently;

Response 3: Thanks to the expert for careful guidance, we modified one by one for the problems pointed out by expert.

Point 4: Paper pictures, tables, formulas, and article layout issues;

Response 4: Due to the confusion caused by the structural change after the article was uploaded, we checked and adjusted the corresponding format.

Point 5: The manuscript should be shortened;

Response 5: We have reduced the content of the article, especially the introduction and section 2.

Reviewer 2 Report

The paper proposes a hungry predation B-spline curve fitting algorithm

with knot guidance.

1. Title is not appropriate.

The paper discusses only about the application of B-spline curves for ship hull design. So the title should include ``ship.''

2. Introduction and Related works

They are too long and just discuss about old papers and it is unclear what are the contributions of the paper. The paper just applies a kind of new technique called hungry predation optimization for B-spline approximation.

3. Figure are dirty.

They are not clear and include noises. They are not a publishable quality.

So my decision is to reject.

Author Response

Response to Reviewer 2 Comments

Point 1: Title is not appropriate.

The paper discusses only about the application of B-spline curves for ship hull design. So the title should include ``ship.''

Response 1: We have revised the title. The new title is B-spline curve fitting of hunger -predation algorithm on ship line design.

Point 2: Introduction and Related works

They are too long and just discuss about old papers and it is unclear what are the contributions of the paper. The paper just applies a kind of new technique called hungry predation optimization for B-spline approximation.

Response2: We have reduced the content of the article, especially the introduction and section 2.

Point 3: Figure are dirty.

They are not clear and include noises. They are not a publishable quality.

Response 3: After the file was uploaded and re-edited, the article layout became messy, we have re-typed the article and modified the figures. Sorry for the inconvenience caused to you for this reason.

Reviewer 3 Report

The paper “A hungry predation optimization B-spline curve fitting algorithm with knot guidance” is devoted to an important issue, but the text requires deep revision.

Abstract: 

Control vertex are the control points ? I suggest unify the paper with control points.  Let us also recall that  the set of control points is called control polygon.

How do you guarantee that are the optimal control points? Is there any theorem? I think that you obtain the control points suitable for obtaining the fitting curve which achieve the accuracy required.

2. Related works

2.1 B-spline theory knowledge.

Please, be more rigorous with the notation. There are serious mistakes. In general, rewrite this section. You may take a look in [53]

As general terms:

Page 2, last paragraph.

The given data are {x_{i}}_{i=0}^{m}, the functions of the B-spline basis are N_{i,k}(x), i=0,\ldots,n, and d_{i}, i=0,\ldots,n, are the control points. I suggest define firstly, the B-spline basis and after the corresponding B-spline curve.

Page 3. Formula (3) is double etiquetted.

Page  3, line 18. Please correct the sentence “For the selection of B-spline degree, in theory, the increase of degree will increase the number of control vertices…” The degree of the B-spline functions is k and the number of control points are n+1.

In general,  rewrite the paragraph and add bibliography that justifies what is being said.

Page 4, Rewrite the last paragraph. For obtaining the control points d_{i} …

p(x) is the fitting curve, curve fitting is the process for obtaining the curve. Please, take into account this in all paper.

In general lines:

The linear system is solved by the least square method since the number of control points (n+1) is minor than the given set of data points (m+1). Moreover, since you want (I think) to obtain the property of endpoint interpolation the first and last control point will coincide with the first and last data point….

2.2 Resarch status

Page 5. Line 5. Rewrite and add bibliography in he sentence “Compared with interpolation, B-spline…”

Page 5. Line 36. Rewrite the sentence: “which causes more control vertices are required to complete …..”.

3. Knot guidance technology.

Page  7 Line 15, According to formula (9) given in [47].

In formula (9) added i=0,…,n.. (please in all paper, added i=0,..n, when necessary).

Please, unify the notation, previously the data points are named q_{i}.

 4. Key parameter settings.

Page 13.

-Figure 7: Please, Replace Figure 7, is doesn't clear. Control polygon is in green or in orange? Original curve is in black or blue?

-Second paragraph: I have various questions.

In the curves obtained in figure 7, the number of control points is 6. What is the degree of the fitting curve?.Please, clarify this in the paper.

Rewrite “fitting result”. Is the fitting curve obtained?.

Page 17: 

Figure 11: Please, replace Figure 11. The colors of the caption don’t be equal.

The data points are not visible in the figure.

I suggest one figure with the fitting face, the fitting curve obtained with method DOM and the fitting curve obtained with HPA.

What is the degree of the fitting curves obtained with all methods?.

Page 19. Figure 14. Please replace the figure, is not clear.  Where  are the six test models?

The function f2(x) is in the same figure thatn f4(x), I suggest appear in other figure.

Page19. Table 3.  I suggest include the number of control points since in the conclusions you are focusing in this.

Page 19. Line -4. Number of knots instead of knot numbers.

Page 20. Table 4. The are references which are wrong.

Page 21. In formula (23) \rho is the degree of the curve. What is the degree of the fitting curves? Doesn't appear in the paper.

Page 22. Please, replace Figure 15 is not clear.

Page 22. Line -11. Rewrite the sentence “It can be seen that when there are…..

Page 24. Figure 18.  What are the fitting error? Are around 10^2?.

5. Discussion and conclusion:

Page 25. Line -6. The sentence “We compared with the typical static knot method…” is repeated.

Please, delate the typical.

Page 25. Line -1. Please clarify what happen about the number of knots. In the paper the number of  knots has been compared.

General questions:

The knots used are uniform in all methods?

What is the degree of the fitting curves obtained in all methods? Are the same?

In your method, the first point and last point of the given data coincide with the first and last control point? In the other methods?

Author Response

Response to Reviewer 3 Comments

Point 1: Control vertex are the control points? I suggest unify the paper with control points.  Let us also recall that the set of control points is called control polygon.

Response 1: Control vertex described in the text are control points. The revised paper has been unified as control points.

Point 2: How do you guarantee that are the optimal control points? Is there any theorem? I think that you obtain the control points suitable for obtaining the fitting curve which achieve the accuracy required.

Response2: The optimal control points refer to the result with the smallest fitness in hunger -predation algorithm when the fitting accuracy requirement is met.

Point 3: Please, be more rigorous with the notation. There are serious mistakes. In general, rewrite section 2.1. You may take a look in [53]

Response 3: This section has been rewritten and the order of the formulas adjusted according to the comments made by the experts.

Point 4: There are some problems with article sentences, formulas, figures.

Response 4: Thanks to the expert for careful guidance, we modified one by one for the problems pointed out by expert. Since the article was re-edited after uploading, resulting in the loss and confusion of the figures, we have rearranged the figures.

Point 5: The knots used are uniform in all methods?

What is the degree of the fitting curves obtained in all methods? Are the same?

Response 5: Knots are not uniform in different methods. In the comparison experiment, we will unify the number of knots, but the values are different, we need to compare the fitting accuracy with the same number of knots. Besides, our method can be applied to curves of various degrees. In practical engineering applications, the cubic curve is a good compromise and is widely used. The degree of curves used in the comparative cases in this paper are all three (order is four).

Point 6: In your method, the first point and last point of the given data coincide with the first and last control point? In the other methods?

Response 6: Solving least squares minimization combined with endpoint constraints comes from the study of Piegl, L and Tiller [17], which is a classic study in B-spline approximation and is widely used. There are also some studies that do not consider endpoint constraints, such as reference[53].

Round 2

Reviewer 1 Report

Some sentences are not written very clearly. For instance, the authors could improve the English level by modifying some parts of the sentences or by replacing some terms with more appropriate words.

For example in the Abstract section the verb to include should be changed ("To solve this problem, this paper proposes a hunger predation optimization B-spline curve fitting algorithm for ship line design (HPA), which includes knot guidance technology, hungry predation optimization technology and adaptive adjustment of algorithm input parameters"). In the Introduction section this sentence can be reformulated or divided into two (In view of the premature loss of population diversity in optimization algorithm, which leads to slow convergence and easy to fall into local optimality, a hunger search strategy is developed to make the hungry individual in the population size appear near the optimal solution more quickly, and the influence of the neighbors in the population size on their position adjustment is further considered.). Likewise check the punctuation, preferably using ; in place of , in all the manuscript to separate indipendent sentences (in sentences like the following one: "HPA achieves the goal of B-spline curve fitting with higher fitting accuracy at the same control points, less control points under the same accuracy requirements, faster operation efficiency, and has better universality, it can better solve practical engineering problems.").

This would result in a more fluent and comprehensible language.

The choice of the tences could also be much correct. See for example, in the Introduction section: "B-spline curve approximation is a hot research topic in the field of computer-aided geometric design in recent years.") where " is " should be replaced by " has been" due to the use of " in recent years".

There are still some errors to be adjusted. See for example the sentences here below:

Abstract

- The ship line often describes by the offset table of discrete data points, which leads to the problems that three view coordinates may not correspond, a large fitting error and the fitted curve cannot be easily modified.

Introduction

- Compared with interpolation, B-spline curve approximation can better reflect the curve shape, but the knots are difficult to determine.

- Now, researchers put higher fitting accuracy at the same knot number [11, 12], using fewer knot vertices under the same accuracy requirements [13, 14] and faster operating efficiency [15, 16] as the main pursuit goals of B-spline curve fitting.

- Therefore, a large population size and iteration number are required, resulting in low efficiency of the algorithm.

Section 2

- For the selection of B-spline degree, the inappropriate degree may lead to results that are difficult to meet the approximation accuracy.

- In 1999, Razdan A [26] considered using the shape information of data points for knot selection

- The nature-inspired optimization algorithm [34-39] is considered to be one of the most practical branches, which can solve complex problems.

Section 3

- This paper mainly focuses on the engineering application of ship line, which mainly include three types of feature points, as shown in the Fig.2.

- population size, half of population still take random values in the interval [0,1], and other half take values in each interval [ui-0.1, ui+0.1], each knot has a 50% chance to mutate, the value interval after mutation is [0,1].

- This paper presents three fitness function that can be applied to HPA.

- In practical engineering application, formulas (18) and (19) are more intuitive and convenient. The smaller fitness can get better fitting effect.

- ..... each individual’s dimension are different accordingly, as shown in Fig.4.

- Instead, we process α, which dimensions are different from ωi.

- then randomly delete a knot according to the probability pi, repeat this process N1 times until the two individual have the same dimension.

Section 4

-....... and every individual's λ are different.

- The more the iteration number, the finer the algorithm searches each time.

- However, when the iteration number is large enough, increasing the iteration number has little effect on the curve fitting accuracy, and will greatly increase the iteration number, which improves the operation time of the program.

- The experimental results show that when E is 0.1, each fitting result uses 2 inner knot, but the fitting accuracy has obvious fluctuations.

- ....... the fitting result are shown in Fig.9.

Section 5

- Besides, DE, WOA and HPA demonstrates stronger search capabilities.

- ......as shown in Fig.14, It can be seen from the figure that the HPA algorithm proposed in this paper has achieved accurate fitting of data points when using 40 knots,

Section 6

- Aiming at the problem that the previous dynamic knot method, the initial knot is randomly selected from the interval [0, 1], which causes many unreasonable knot vectors.

Author Response

Point 1: Some sentences are not written very clearly. For instance, the authors could improve the English level by modifying some parts of the sentences or by replacing some terms with more appropriate words.

Response 1: We have corrected the sentence problem you pointed out and checked the full text. Thank you sincerely for your patience in pointing out the problematic sentences one by one. Our native language is not English, so some sentences are not clear enough. Modified sentences are highlight in yellow. Thanks again.

Reviewer 2 Report

The paper is well revised according to the reviewers' comments and my decision to to accept as it is.

Author Response

Point 1: The paper is well revised according to the reviewers' comments and my decision to to accept as it is.

Response 1: Thanks very much for taking your time to review this manuscript. I really appreciate all your comments and suggestions. It has important guiding significance for this manuscript.

Reviewer 3 Report

The authors have greatly improved the manuscript and corrected the mistakes.

Before the publication I would like to point out that the authors reconsider in the abstract "the optimal control points". For being the optimal control points you may be sure that there exists a theorem satisfying that. The algorithm finds a set of control points but maybe we can find  another set of control points for which we can achieve the same accuracy (or better), therefore maybe it is not the set of control points found by the algorithm the optimal set one.

Author Response

Point 1: Before the publication I would like to point out that the authors reconsider in the abstract "the optimal control points". For being the optimal control points you may be sure that there exists a theorem satisfying that. The algorithm finds a set of control points but maybe we can find another set of control points for which we can achieve the same accuracy (or better), therefore maybe it is not the set of control points found by the algorithm the optimal set one.

Response 1: Thanks very much for taking your time to review this manuscript. I really appreciate all your comments and suggestions. The theorem for determining the optimal control point is still a question to be considered. In this paper, we refer to the result with the best fitness in the population as the optimal solution. The optimal solution found by the optimization algorithm belongs to the local optimum. The fitting results of each HPA are different, so we propose an adaptive parameter adjustment method. While ensuring high operating efficiency, it ensures that the optimal solution obtained by each calculation is close to the global optimal solution.